# The Role of Red Cell Distribution Width as a Prognostic Marker in Chronic Liver Disease: A Literature Review

**DOI:** 10.3390/ijms24043487

**Published:** 2023-02-09

**Authors:** Hunain Aslam, Fouzia Oza, Khalid Ahmed, Jonathan Kopel, Mark M. Aloysius, Aman Ali, Dushyant Singh Dahiya, Muhammad Aziz, Abhilash Perisetti, Hemant Goyal

**Affiliations:** 1The Wright Center for Graduate Medical Education, 501 S. Washington Ave, Scranton, PA 18505, USA; 2Department of Medicine, The Wright Center for Graduate Medical Education, 501 S. Washington Ave, Scranton, PA 18505, USA; 3Department of Internal Medicine, Texas Tech University Health Sciences Center, Lubbock, TX 79430, USA; 4Department of Internal Medicine, Central Michigan University College of Medicine, Saginaw, MI 48603, USA; 5Department of Gastroenterology and Hepatology, University of Toledo Medical Center, Toledo, OH 43614, USA; 6Department of Gastroenterology and Hepatology, Kansas City VA Medical Center, Kansas City, KS 64128, USA; 7Center for Interventional Gastroenterology at UT (iGUT), Division of Gastroenterology, Hepatology, and Nutrition, The University of Texas Health Science Center, 6431 Fannin, MSB 4.234, Houston, TX 77030, USA

**Keywords:** red cell distribution width, viral hepatitis, non-alcoholic fatty liver disease, autoimmune hepatitis, primary biliary cholangitis, hepatocellular carcinoma

## Abstract

Liver disease is one of the leading public health problems faced by healthcare practitioners regularly. As such, there has been a search for an inexpensive, readily available, non-invasive marker to aid in monitoring and prognosticating hepatic disorders. Recently, red blood cell distribution width (RDW) has been found to be associated with various inflammatory conditions with implications for its use as a potential marker for assessing disease progression and prognosis in multiple conditions. Multiple factors effect red blood cell production whereby a dysfunction in any process can lead to anisocytosis. Furthermore, a chronic inflammatory state leads to increased oxidative stress and produces inflammatory cytokines causing dysregulation and increased intracellular uptake and use of both iron and vitamin B12, which leads to a reduction in erythropoiesis causing an increase in RDW. This literature review reviews in-depth pathophysiology that may lead to an increase in RDW and its potential correlation with chronic liver diseases, including hepatitis B, hepatitis C, hepatitis E, non-alcoholic fatty liver disease, autoimmune hepatitis, primary biliary cirrhosis, and hepatocellular carcinoma. In our review, we examine the use of RDW as a prognostic and predictive marker for hepatic injury and chronic liver disease.

## 1. Introduction

Chronic liver diseases (CLD) are a leading cause of morbidity and mortality in the United States (US). According to the Centers for Disease Control (CDC), 4.5 million individuals had active liver disease in the US alone in 2018 [1]. Furthermore, it is estimated that liver disease accounts for approximately 2 million deaths annually worldwide [2]. Chronic liver diseases, such as those caused by obesity, alcoholism, viral hepatitis, and autoimmune liver disease, can lead to chronic inflammation that results in fibrosis, cirrhosis, and hepatocellular carcinoma (HCC). Standard methods for monitoring liver function include albumin, prothrombin (PT), the international normalized ratio (INR), platelet count, serum alanine aminotransferase level (ALT), aspartate transaminase (AST), alkaline phosphatase (ALP), bilirubin, gamma-glutamyl transpeptidase (GGT), and lactate dehydrogenase (LDH). Imaging modalities, such as abdominal magnetic resonance imaging (MRI), ultrasound, and computed tomography (CT) scans, are routinely performed to screen and follow the response of therapy for HCC. However, these imaging modalities have limitations. For example, patients with pre-existing conditions such as underlying chronic kidney disease (CKD) with a severely low glomerular filtration rate (GFR) cannot undergo MRI or CT scans with contrast. The presence of non-compatible metal prostheses, metal stents, pacemakers, or certain artificial heart valves also restricts MRI use, leading to a search for a potential biomarker that is inexpensive, non-invasive, and has diagnostic and prognostic capabilities to augment the predictive value of current liver function tests.

Red blood cell distribution width (RDW), a commonly performed laboratory parameter as part of complete blood count (CBC), measures the variation in red blood cell (RBC) size. Red blood cell distribution width is obtained by measuring the standard deviation in RBC volume and erythrocyte volume for the mean corpuscular volume (MCV). A normal RDW ranges between 11–15% [3]. A low RDW value is clinically insignificant; however, a high value may be associated with elevated inflammatory markers, such as estimated sedimentation rate (ESR) or C-reactive protein (CRP), or interleukin (IL) [4]. In addition, many studies have shown that RDW could be used to predict prognosis in a multitude of gastrointestinal disorders, including esophageal squamous cell carcinoma, autoimmune gastritis, gastric cancer, celiac disease, colon carcinoma, intestinal tuberculosis, inflammatory bowel disease, and liver disorders [5,6,7,8,9,10,11]. Therefore, many studies have sought to delineate any correlation between RDW and the prognosis/mortality of various hepatic diseases.

Multiple pathological factors play a role in RDW variation, particularly diseases that cause continuous injury to the liver parenchyma. The resulting injury to the liver parenchyma leads to a chronic inflammatory state, which causes an increase in circulating proinflammatory cytokines, such as TNF-α, IL-1, IL-6, IL-8, IL-12, IL-18, and IFN-γ. These cytokines are known to interfere with hamper erythropoietin production and iron availability. Of the cytokines listed, IL-6 causes a decrease in duodenal iron uptake with a corresponding increase in iron uptake in macrophages, which ultimately leads to reduced functional iron availability for effective erythropoiesis [12]. Furthermore, chronic inflammation leads to increased reactive oxygen species, free radical formation, and inflammatory markers causing impaired iron metabolism and depletion of antioxidant levels. This resulting increase in inflammation causes a cascade of reactions that leads to alterations in RBC morphology and apoptosis. Furthermore, nutritional deficiencies and cachexic states can lead to the depletion of minerals and vitamins, such as iron, folate, and vitamin B12, from the body [13]. These factors cumulatively play a vital role in regulating inflammation and erythropoiesis, leading to anisocytosis and increased RDW. Chronic inflammation also causes fibrosis and cirrhosis, which can lead to cancer development if left uncontrolled [14]. As RDW is frequently elevated in inflammatory and oxidative stress states [15], it is known to correlate to disease severity in a variety of liver disorders, including hepatitis B (HepB), hepatitis C (HepC), non-alcoholic fatty liver disease (NAFLD), primary biliary cholangitis (PBC), and HCC. In this article, we examine the literature regarding the prognostic value of RDW in assessing various liver disorders while summarizing its putative mechanisms (Table 1).

## 2. Hepatitis B

Hepatitis B is one of the most common causes of chronic hepatitis and end-stage liver disease, affecting 240 million individuals globally [31]. If left untreated, HepB causes acute liver failure, cirrhosis, and HCC in up to 40% of patients [32]. The pathophysiology of liver disease in HBV infection is primarily immune mediated, while HBV can occasionally directly injure the liver via cytotoxicity [32]. HepB-infected cells are more likely to be lysed by T lymphocytes when exposed to HBsAg and other nucleocapsid proteins found on cell membranes. The bulk of HepB DNA is removed from the hepatic system before maximal T-cell infiltration, indicating that the immune response is probably more active in the early stages of infection [32]. The cytotoxic T-cell response to HepB-infected hepatocytes is relatively ineffectual [32]. The immune response might not be the only cause of liver damage in people with HepB [32]. Patients with hepatitis B who have undergone liver transplants and are receiving immunosuppressant medication may also experience HepB-associated damage [32].

Hepatitis B treatment aims to achieve sustained suppression of viral replication, thereby preventing fibrosis, cirrhosis, and eventual progression to HCC [33]. These patients frequently require close follow-up and monitoring using several laboratory markers, including viral load, serum hepatitis B surface antigen (HBsAg), and ALT, to determine the disease prognosis, treatment efficacy, and progression of Hep [34]. Red cell distribution width is a laboratory parameter recently studied as a marker for inflammation and fibrosis in patients with chronic HepB (CHB) infection (Figure 1). Historically, RDW was used as a marker for anemia; however, chronic inflammatory conditions, such as HepB, can alter RDW. Increased inflammatory responses lead to increased intracellular uptake and usage of iron and vitamin B12 as well as increased production of inflammatory cytokines and free radicals coupled that deplete antioxidants. Together, this leads to impaired erythropoiesis and the production of larger dysfunctional RBC, which leads to an increase in RDW (Figure 2). This section will highlight a few studies that have analyzed the significance of such an inflammatory response and its effect on RDW [12,35].

Lou Y. et al. [16] conducted a study in which they found that RDW values were increased in patients with HepB and were associated with disease severity. A total cohort of 171 individuals was used in the analysis, consisting of healthy controls and acute HepB, CHB, and chronic severe HepB patients. The RDW values for the control and acute HepB, CHB, and chronic severe HepB patients were 13.03 (11.7–14.36), 14.38 (95% CI 12.66–16.1, *p* < 0.05), 16.37 (95% CI 13.94–18.8, *p* < 0.001), and 18.30 (95% CI 15.19–21.41, *p* < 0.001), respectively [16]. The study found that the RDW values were significantly elevated in patients with CHB and chronic severe hepatitis B infection [16].

A meta-analysis was performed by Fan X. et al. [22] in which 24 studies with a total of 5481 subjects were analyzed, of whom 2209 were healthy controls and 3272 were HBV-infected patients. Of the 1768 cases of CHB, 256 were cases of acute on chronic liver failure (ACLF) and 1247 were cases of cirrhosis [22]. Their analysis showed that RDW levels were significantly higher in CHB patients compared to the healthy controls as well as in patients with ACLF [22]. The authors also compared RDW values between HepB-related cirrhosis and CHB. Using a random effect model, their analysis showed cirrhotic HepB patients had higher RDW levels than CHB patients [22]. However, Fan X. et al. did not find a statistically significant difference in RDW in patients with ACLF and HBV cirrhosis. Therefore, Fan X. et al. concluded that RDW might differentiate chronic HBV from healthy controls, ACLF, and cirrhosis from CHB [22].

Other studies have been conducted to find an association between increased RDW and HepB in various stages of hepatitis infection. A retrospective study by Chen B. et al. [17] examined 458 patients with liver biopsy diagnosed with CHB. Chen B. et al. analyzed the significance of RDW in the late vs. early stages of fibrosis and cirrhosis among patients with CHB [17]. Their results for multivariate analysis of fibrosis showed RDW to be significantly associated with the severity of disease at an odds ratio (OR) of 1.80 (95% CI 1.2–2.69, *p* = 0.004) [17]. Similar results were shown for RDW and the stage of cirrhosis at OR 2.06 (95% CI 1.21–3.51, *p* < 0.001) [17]. Chen B. et al. also found that RDW was positively correlated with significant fibrosis and cirrhosis [17]. Another study by Karagoz E. et al. [18] found RDW to be independently associated with fibrosis stages with a receiver operating curve (ROC) value of 0.672. In addition, RDW was associated with prognostic specificity of 42.5% and a sensitivity of 91.5%.

A subsequent study by Humane R. et al. [19] showed that RDW correlates with different liver injury stages in hepatitis B. Humane R. et al. [19] compared HepB-related liver cirrhosis, chronic hepB, and a healthy control. They found RDW to be an independent predictor of HepB-related liver cirrhosis. Another study by Xu W. et al. [20] conducted a retrospective analysis where levels of inflammatory activity and RDW were analyzed in HepB patients. The resulting multivariate analysis was significant for RDW at an OR 1.146. Specifically, Xu W. et al. found that a 1% increase in RDW value was associated with a 25.9% increase in the risk for advanced inflammatory activity for CHB [20]. Furthermore, Zu W. et al. found that RDW was an independent predictor for fibrosis in an adjusted analysis with an OR 1.121 (95% CI 1.002–1.253, *p* = 0.04) [20]. Additionally, Wang H. et al. [21] found a strong correlation between RDW, with it being a predictor of severity of both liver inflammation (OR 1.660, 95% CI 1.284–2.148, *p* < 0.001) and liver fibrosis (OR 1.487, 95% CI 1.204–1.836, *p* < 0.001). Together, these studies suggest an association between RDW and the severity of CHB infection that could be utilized as an independent predictor and prognostic marker for liver inflammation and fibrosis. Despite this, more research is needed to delineate the exact mechanisms of RDW change in HepB.

## 3. Hepatitis C

Hepatitis C affects around 170 million people worldwide. Along with HepB, HepC is a leading cause of cirrhosis, liver failure, and HCC [36]. Hepatitis C is a positive-strand ribonucleic acid (RNA) virus that is spherical, enclosed, and is around 55 nm in diameter. Although it belongs to the Flaviviridae family, the hepacivirus is unique enough to be categorized as a different genus [36]. The length of the genome is roughly 9.6 kb. It produces at least ten proteins after first encoding a polyprotein [36]. These include two proteins required for virion formation (p7 and NS2), two structural proteins—the nucleocapsid protein and a core (C)—and two envelope proteins [36]. The NS5B RNA polymerase exhibits a very high virion turnover in the absence of proofreading [36]. At least four co-receptor molecules allow the HepC RNA virus to enter the hepatocyte through endocytosis. Its positive-stranded RNA is internalized in the cytoplasm, where it is uncoated and translated into mature peptides [36]. These are then broken by virally encoded NS3-4a serine proteases as well as host proteases [36]. These fully developed peptides then move onto the endoplasmic reticulum, forming a replication complex with RNA-dependent RNA polymerase [36]. This enzyme converts the positive RNA strand into its intermediate negative strand, which then acts as a template for producing fresh positive strands [36]. These become mature virions combined with a core and envelop glycoprotein and then leave the cell through exocytosis. The host’s genome cannot be integrated by HepC [36].

Hepatocellular carcinoma is associated with severe liver fibrosis and cirrhosis secondary to chronic liver inflammation. It is believed that chronic inflammatory conditions, increased oxidative stress, nutritional deficiencies, and HepC-treatment-induced hemolytic anemia [37] all play a significant role in causing ineffective erythropoiesis, leading to larger immature erythrocytes and causing an increase in the RDW value in HepC patients. However, due to the complexity of the disease process and the limited available literature, the exact mechanism of action and the specificity and sensitivity of RDW in HepC patients is not available. HepC causes ineffective erythropoiesis by increasing oxidative stress and nutritional deficiencies, which leads to hemolytic anemia [38]. A liver biopsy is the most accurate method for monitoring the progressing of liver damage from HepC [38]. However, a liver biopsy is invasive and expensive [38]. As such, many serum markers are utilized to help assess liver fibrosis, including the fibrosis-4 score (FIB-4), the aspartate transaminase to platelet count ratio index (APRI), and the NAFLD fibrosis score (NFS), as well as imaging to evaluate the degree of fibrosis in patients with hepatitis C [39]. In recent years, RDW has been proposed as another serum marker that can be used to assess liver fibrosis and the progression of HepC [40].

In a retrospective analysis carried out by He Q. et al. [41], a total of 94 HepC-infected patients (52 chronic HepC (CHC) patients and 42 HepC-related cirrhosis patients) and 84 control patients were examined to measure their RDW levels. Different demographic variables were compared, and RDW was elevated in the HepC group (SD 15.18 ± 2.59) compared to the control group (SD 13.39 ± 1.17). Furthermore, RDW was significantly increased in HepC cirrhosis (16.44 ± 3.01) compared to CHC (14.12 ± 1.54). RDW was also significantly elevated in HepC with a higher risk factor for cirrhosis (OR 1.494, 95% CI 1.036–2.155, *p* = 0.032). An ROC was constructed to assess the diagnostic capabilities of markers in identifying cirrhosis in CHC patients, which showed that the area under the curve (AUC) for RDW was found to be 0.791 ± 0.045. However, the AUC for the RDW platelet ratio (RPR) was more significant at 0.960 ± 0.018. The authors concluded that RDW and RPR can be considered along with other biomarkers to indicate the severity of HepC infection in patients and can be used as a fast and inexpensive marker in HepC patients. Similarly, Nar R. et al. [24] performed a retrospective study that also showed that RDW was elevated in the HepC patients compared to the control group (15.05 ± 1.88 vs. 13.76 ± 1.05, *p* < 0.001).

## 4. Hepatitis E

Hepatitis E (HepE) is another viral hepatitis usually transmitted through the fecal–oral route. According to the WHO, there are 20 million HEV infections worldwide, leading to around 3.3 million symptomatic cases annually [42]. Approximately 44,000 people died from HepA, accounting for 3.3% of mortality due to viral hepatitis in 2015 [42]. The icosahedral, non-enveloped, single-stranded RNA HepE has a diameter between 27 and 34 nm. HEV has been classified into four genotypes, 1–4 [36]. Human viruses of genotypes 1 and 2 are primarily present in underdeveloped nations in Africa, Asia, Central America, and the Middle East. They are spread via the fecal–oral route via polluted water [36]. In affluent nations such as the United States, Australia, Japan, and China, genotypes 3 and 4 are mostly found in animals (zoonotic) and are spread to people by eating undercooked meat, such as pork and beef. Hepatitis E infections might take anywhere between 28 and 40 days to develop [36]. After being ingested, HepE enters the portal circulation through the gastrointestinal mucosa and travels to the liver [36]. Other than in the liver, no other tissues have shown evidence of HepE replication [36]. Although HepE can cause morphologic changes in the liver that resemble both cholestatic and traditional acute hepatitis, these characteristics are not a reliable indicator of the presence of the virus [36].

In the majority of HepE infections, patients develop mild symptoms; however, in rare cases, acute HepE infection can lead to severe fulminant hepatic failure, particularly in pregnant females during the third trimester [43]. Hepatitis E also causes CLD leading to ACLF with high mortality of 0–67% [44]. Chronic HepE (CHE) infection has been detected in immunosuppressed patients, such as solid-organ transplant recipients, patients with HIV, or patients undergoing chemotherapy. Hepatitis E is detected by blood anti-HEV IgM and IgG antibodies with monitoring of ALT, AST, and bilirubin levels. It is believed the hepatitis E may influence RDW levels through systemic inflammation and bone marrow suppression which leads to anemia in HepE-infected patients. Those with liver failure tend to have elevations in RDW. Due to the scarcity of available literature, further studies need to be carried out to establish the exact cause of elevated RDW in HepE patients.

A meta-analysis performed by Jian W. et al. [25] inclusive of 302 HepE patients with and without liver failure and a control compared the RDW values in each group. The authors found significant RDW elevations between the control and the HepE (with/without) liver failure group as well as the HepE without liver failure group and the HepE with liver failure group [25]. An AUC was performed to assess RDW’s significance in predicting HepE-related liver failure, which showed an AUC of 0.63 with a sensitivity of 0.58 [25]. In addition, 62.5% of the subjects in the HepE with liver failure group were found to have an RDW > 14.8 with a sensitivity of 0.63 and specificity of 0.67 [25]. The investigators further analyzed the data by correlating RDW with serum albumin concentration (ALB), total bilirubin (TBIL), and Child–Pugh score (CPS); RDW was found to be positively correlated with TBIL (r = 0.392; *p* < 0.001) and CPS (r = 0.385; *p* < 0.001). A negative correlation between ALB and RDW was also found in the study (r = −0.47 at *p* < 0.001) [25]. This study evaluated the role of RDW in predicting the development and prognosis of liver failure following HepE infection and found that RDW can be used as a preliminary diagnostic marker for liver failure in HepE patients.

## 5. Non-Alcoholic Fatty Liver Disease

Non-alcoholic fatty liver disease (NAFLD) is a spectrum of liver disorders, including Non-alcoholic fatty liver and non-alcoholic steatohepatitis (NASH) [45]. Non-alcoholic fatty liver disease has been linked to obesity, diabetes, dyslipidemias, insulin resistance, and metabolic syndrome [45]. Additionally, it has been demonstrated that inorganic arsenic exposure and the onset of NAFLD, which is indicated by increased alanine transferase, are related in time (ALT) [45]. NAFLD correlates with cardiovascular risk factors because of its strong relationship to metabolic syndrome, which, along with end-stage liver cirrhosis and hepatocellular carcinoma, is another determinant in these patients’ death [45]. Non-alcoholic fatty liver disease consists of hepatic steatosis without evidence of hepatocyte ballooning, and NASH consists of hepatocyte injury (ballooning) in addition to the findings of NAFL.

Pathogenesis of NAFLD consists of steatosis, lipotoxicity, and inflammation [46]. It has the potential risk of progression to end-stage liver disease, including fibrosis, cirrhosis, or HCC. Under normal physiologic conditions, the liver metabolizes free radicals and ROS, which are pivotal in producing antioxidants and counteracting stress. However, in a compromised state, it cannot do so. Specifically, NAFLD develops and progresses as a result of both environmental and genetic factors [47,48,49]. Patients with NAFLD’s first-degree relatives are at higher risk than the overall population. The sirtuin or cAMP-responsive element-binding protein H (CREBH) or histone amino-terminal ends of chromatin are maintained by histones (SIRT1) [47,48,49]. Genetic research has demonstrated that activation of SIRT1 is believed to contribute to the development of NAFLD [47,48,49]. The step of NAFLD is brought on by insulin resistance, which causes triglycerides to condense into fat droplets that build up in the cytoplasm of hepatocytes and cause steatosis [47,48,49]. Furthermore, excessive carbohydrates stimulate fatty acid production by the liver [47,48,49]. The liver is more susceptible to damage when it contains excessive fatty acids [47,48,49]. It is thought that the damage is brought on by peroxisomal fatty acid oxidation, mitochondrial respiratory chain formation of reactive oxygen species (ROS), fatty acid metabolism by cytochrome P450, and hepatic metabolism of alcohol obtained from the gut [47,48,49]. Obesity also contributes to the second hit as adipose tissue releases inflammatory mediators such as leptin, tumor necrosis factor (TNF)-alpha, and interleukin (IL)-6, which damage hepatocytes [47,48,49].

It has been hypothesized that a chronic inflammatory state and an increased oxidative state lead to the production of irregular-sized RBCs, leading to increased RDW. The studies above show RDW to be significantly associated with liver injury in NAFLD. Due to an association between inflammation and oxidative stress of NAFLD progression, studies have been performed to establish a possible link between RDW and the degree of fibrosis in NAFLD.

The association of RDW to NAFLD was evaluated by Yang W. et al. [27] in a retrospective study of 2256 patients, of whom 619 were found to have NAFLD. Yang W. et al. [27] found that patients with NAFLD were more likely to have high levels of RDW than those without NAFLD (13.23 ± 1.01 and 12.96 ± 1.08, respectively). Cengiz et al. [26] also found that patients with NASH had higher RDW values than patients with only steatosis and healthy controls. Cengiz et al. [26] reported an increase in RDW among patients with advanced fibrosis compared to mild fibrosis. Similarly, Dogan S. et al. [50] found that the specificity and sensitivity of using RDW as an indicator for patients with NASH to be 73.3% and 79.5%, respectively.

Kim H.M. et al. [51] evaluated the correlation between RDW and the level of fibrosis score using the BARD score and Fibrosis 4 (FIB-4) score. The BARD score comprises three variables: BMI > equal to 25, AST/ALT (AAR) ratio > equal to 0.8, and type 2 diabetes. The fib-4 score is calculated with the formula: age (years) × AST [IU/L]/platelet count [expressed as platelets × 109/L] × (ALT [IU/L]). The elevated RDW values correlated with both the BARD score and Fib-4 score: 12.59 ± 0.62 (BARD score 0,1), RDW 12.99 ± 0.85 Bard (2–4) (*p* < 0.001). RDW 12.61 ± 0.77 for a FIB-4 score < 1.3, and RDW 12.89 ± 0.71 for a FIB-4 score greater than or equal to 1.3. This study concluded that RDW is independently associated with advanced fibrosis in patients with NAFLD. The findings were adjusted for age, sex, Hb, MCV, hsCRP, history of smoking, history of diabetes, and history of hypertension.

## 6. Autoimmune Hepatitis

Autoimmune hepatitis (AIH) is a chronic disease caused by immune-mediated inflammation of the liver leading to liver fibrosis and cirrhosis [52,53]. The majority of patients (60%) have chronic hepatitis but lack serologic proof of viral infection [52,53]. Anti-smooth muscle autoantibodies are also linked to the pathogenesis of AIH [52,53]. The current hypothesis for pathogenesis postulates that a genetically predisposed person’s failure to develop immunological tolerance results in T-cell-mediated inflammation brought on by a variety of environmental stressors [52,53]. Toxins, medicines, and infections are common triggers. Several human leukocyte antigen (HLA) haplotypes are more prone to autoimmune hepatitis development [52,53]. Various ethnic groups have different susceptible alleles. Susceptible alleles are found on the short arm of chromosome 6, notably in the area of DRB-1, in White North Americans and Northern Europeans [52,53]. Nitrofurantoin and minocycline are widely known for causing autoimmune hepatitis [52,53]. Drugs that inhibit tumor necrosis factor-alpha have more recently been connected to autoimmune hepatitis [52,53].

Autoimmune hepatitis can be triggered by environmental factors, immune imbalances, or drugs. AIH is currently diagnosed by a scoring system that comprises serum transaminases and IgG levels, auto-antibodies titer, the presence of histological features on biopsy, and the absence of viral markers of hepatitis. Autoimmune hepatitis is a severe and chronic disease historically diagnosed with liver biopsy, an invasive procedure. Multiple non-invasive indices are evaluated using the liver enzymes ALT, AST, and IgG. However, RDW has shown results as a non-invasive marker to predict and diagnose AIH along with other markers and could be used as a possible prognosis marker in the future. The exact mechanism is unknown; however, inflammatory conditions in the body affect iron metabolism and hematopoiesis, leading to increased RDW levels. Recently several studies have found a correlation between RDW and autoimmune diseases [9] and RDW and CLD.

A retrospective cross-sectional study carried out by Wang H. et al. [29] analyzed 92 patients with liver-biopsy-confirmed AIH and compared RDW, ALT, and IgG to the degree of inflammation. Depending on the severity, the degree of inflammation was graded as minimal (G1), mild (G2), moderate (G3), or severe (G4). Their analysis revealed that the RDW in AIH patients with severe inflammation was more significantly elevated than in mild inflammation. However, there was no significant association between ALT and IgG. A significant positive correlation was noted between the degree of inflammation and RDW and IgG. At the same time, AST showed no significant correlation with RDW. An ROC was constructed to evaluate the significance of RDW in identifying liver inflammation in AIH patients. The analysis showed that the ROC for RDW was better at predicting liver inflammation (0.739, 95% CI 0.634–0.843) compared with ALT (0.496, 95% CI: 0.370–0.622, *p* = 0.003) and IgG (0.594, 95% CI: 0.473–0.714, *p* = 0.049), with a sensitivity of 67.8% and a specificity of 75.76%. Wang H. et al.’s study concluded that RDW showed promising indications for predicting significant liver inflammation in AIH patients [29].

Another study by Zeng et al. assessed non-invasive markers to monitor the AIH disease course using RDW [28]. A total of 151 individuals were enrolled of whom 76 were diagnosed with AIH and they were paired with 75 healthy individuals. The autoimmune hepatitis patients were divided into the chronic AIH group and the AIH cirrhosis group. Many indicators, including RDW, were tested, revealing that RDW levels were higher in AIH individuals (14.60) compared to the healthy controls (13.00). In addition, RDW levels were significantly elevated in the AIH cirrhosis group at 15.75 compared to the chronic AIH group at 13.85. A univariate analysis was additionally performed and resulted in an RDW with AN OR at 1.447 (95% CI: 1.148–1.824). Similarly, multivariate analysis showed an RDW with AN adjusted OR at 1.414 (95% CI: 1.086–1.842).

## 7. Primary Biliary Cholangitis

Primary biliary cholangitis (PBC) is a chronic and progressive immune-mediated disease caused by the autoimmune destruction of intrahepatic bile ducts [54,55,56]. Primary biliary cholangitis is an autoimmune condition that is believed to result from both environmental triggers and genetics [54,55,56]. First-degree relatives of the index patient have a 100-fold higher occurrence of the condition [54,55,56]. Various studies have shown that environmental triggers such as xenobiotics, toxic waste sites, reproductive hormone replacement therapy, nail polish, and urinary tract infections are linked in animal models [54,55,56]. Chemical and environmental elements are believed to injure the tissue directly, causing inflammation [54,55,56]. Genetic predisposition and environmental stimuli are assumed to be involved in the development of PBC [57,58]. Substantial illness prevalence in first-degree relatives, with an odds ratio of 11, suggests a hereditary susceptibility [57,58]. In monozygotic twins, there is also a high level of concordance [57,58]. The relative risk for PBC development is higher in index women’s daughters [57,58]. Primary biliary cirrhosis has been linked to a number of human leukocyte antigen (HLA) alleles, including DRB1, DR3, DPB1, DQA1, and DQB1. HLA-DRB1*11 has been found to be protective, but HLA-DRB1*08 is frequent in people of European and Asian heritage [57,58]. Toxic waste, smoking, nail polish, hair dye, and different xenobiotics are among the environmental triggers (e.g., *Escherichia coli*, *Mycobacterium gordonae*, and *Novosphingobium aromaticivorans*) [57,58]. The presence of a humoral and cellular response to an intracytoplasmic antigen, the presence of anti-mitochondrial antibodies, and the involvement of T lymphocytes in the destruction of bile ducts are all signs that these environmental triggers cause an autoimmune reaction in patients who are genetically susceptible [57,58]. Additionally, lipoylated protein-containing bacteria trigger an immune response that uses molecular mimicry to target their own lipoylated proteins [57,58]. The attachment of a glutathione residue blocks the exposed epitope during apoptosis in somatic cells [57,58].

The occurrence of PBC is more commonly found in women over the age of 35. Historically PBC has been diagnosed via various blood markers, and the gold standard of diagnosis has been liver biopsy. However, no marker is currently present to detect the stages of liver inflammation or liver fibrosis and to evaluate the treatment response. Hence, several biological and clinical markers have been assessed as potential prognostic markers; one such marker is RDW. This section reviews the studies carried out on the association between RDW and PBC. In a retrospective analysis performed by Wang H. et al. [30], a total of 93 patients with liver-biopsy-confirmed PBC were enrolled and divided into early stage (Stage I) and advanced stage (Stages II–IV). Different markers were analyzed and compared between the early and advanced stages. A ROC showed an RDW AUC (0.66, 95% CI 0.54–0.79) at *p* = 0.019 and a specificity of 92.9%. Furthermore, RDW was significantly elevated in the advanced stages (14.4, 95% CI 13.3–15.3) compared to the early stages (13.6, 95% CI 12.9–14.4). Wang H. et al. concluded that RDW and RPR could provide essential information for predicting the severity of PBC [30]. The results of this study are consistent and similar to our previously mentioned studies with the probable mechanism of action. However, minimal literature is currently available on PBC patients.

## 8. Hepatocellular Carcinoma

Hepatocellular carcinoma (HCC) is one of the most common malignancies and one of the most frequent primary liver malignancies. It ranks 5th in men and is the second most lethal solid tumor [59]. HCC is usually caused by liver fibrosis and cirrhosis, and it is usually the final sequelae of chronic liver inflammation caused by CHB, CHC, alcoholic fatty liver disease and NAFLD, and other liver diseases. In recent years, RDW levels have been suggested as a helpful, predictive, and prognostic marker for patients with HCC [60]. Though the exact mechanism of RDW and HCC has not been established, it is hypothesized that RDW could be increased due to inflammation similar to CRP levels. Inflammation usually leads to ineffective erythropoiesis, which can lead to defective RBC production and cause an increase in RDW. A prospective study by Zhao T. et al. [61] analyzed 106 patients with HCC over five years, evaluating pre-operative RDW value and clinicopathological characteristics. The patients were divided into groups: those with an RDW higher than 14.5 and those with an RDW lower than 14.5. In patients with a higher RDW, a significant difference was found in tumor stage, tumor size, and vascular invasion. Similarly, 1-, 3- and 5-year survival rates were worse in the higher RDW group at 67.9%, 10.0%, and 0% compared to the lower RDW group at 69.7%, 32.4%, and 18.1%, respectively.

Another study was performed by Smirne C. et al. [62], in which a total of 314 patients were analyzed. The patients were divided into training and validation cohorts. In the training group, 208 patients were retrospectively enrolled, while 106 patients were prospectively enrolled in the validation group. After which, their liver disease staging was stratified as per the Barcelona clinic liver cancer staging system. Each cohort divided the patients into two different RDW groups: one with an RDW ≤ 14.6 and the other with an RDW > 14.6. The median RDW value in the training cohort was 14.6, in which 54.6% had an RDW < 14.6, while in the validation cohort, the median value was 15.3, with 60.4% having an RDW > 14.6. The median survival in the training group was higher in the RDW <14.6 group vs. >14.6 group with (HR 0.43, 95% CI 0.31–0.60, *p* < 0.0001). Similarly, in the validation cohort, median survival was greater in the RDW <14.6 group compared to the others (HR 0.28, CI 0.17–0.47, *p* < 0.0001). Similarly, the 1-, 3- and 5-year survival rates with an RDW ≤ 14.6 were 79%, 57%, and 42%, compared to rates for an RDW > 14.6 which were 48%, 29%, and 18%. The OR of dying within the first year from diagnosis with an RDW >14.6 vs. ≤14.6 was 4.2. Furthermore, a COX proportional hazard analysis revealed RDW to be an independent predictor of survival in both cohorts with a hazard ratio (HR) of 1.13 (CI 1.06–1.21, *p* = 0.0003).

## 9. Future Directions

Red cell distribution width could potentially be key to better understanding and managing chronic liver diseases and other chronic medical conditions. RDW assessment is less expensive than other biomarkers and does not require intrusive procedures such as biopsies [63]. Therefore, further study is needed to explore the potential of RDW as a reliable prognostic biomarker for emerging viruses and to develop methods to reduce the drawbacks of employing this readily available predictive measure [63]. In addition, understanding the exact pathophysiological changes leading to an increase in RDW could help identify confounders and identify possible disease pathways that could be altered to possibly change disease outcomes and allow for better management of chronic medical conditions. Hence, future studies using random clinical trials should be undertaken to determine RDW’s exact sensitivity and specificity in terms of being a prognostic and predictive marker in chronic liver diseases.

However, there are notable limitations to using RDW for both diagnostic and prognostic purposes regarding liver diseases. RDW values have been shown to be impacted by several factors, such as malnutrition, bone marrow depression, erythropoietin use, thyroid dysfunction, iron or vitamin B12 deficiency, and cardiovascular disease [63]. These factors lessen RDW specificity as a prognostic indicator for disease development [63]. In addition, age is a complicated issue in applying the biomarker for prognostic prediction of viral illnesses because RDW also rises with age. Given the multitude of variables linked to higher RDW, it might be difficult to account for every potential confounding variable in a single study [63]. Additionally, RDW changes over the course of many infectious diseases, including viral diseases. As a result, using RDW measurement as a prognosis predictor may overlook dynamic changes at various stages of disease progression [63]. The specificity of RDW as a predictor of disease prognosis is also constrained by abnormal pathogenic mechanisms that often occur in the blood circulation.

As such, the International Council for Standardization in Hematology should first advocate for the standardization of RBC distribution curve analysis as well as the selection of anticoagulants and standard-deviation- or coefficient-variation-based calculations [63]. Second, establishing age- and population-specific reference ranges will be crucial for global geographical regions and subregions because RDW varies between populations and different age groups [63]. Third, a predictive model that considers several variables that may impact RDW and assess each variable’s contributions will be a step toward overcoming RDW’s biological limitations as a prognostic biomarker for viral illnesses [63]. Otherwise, when employing RDW as a predictive biomarker for disease progression or death, age and other confounding factors should be taken into account in multivariate analysis [63]. Last but not least, recommendations on optimal timing and conditions for sample collection should be put in place because they have an impact on RDW levels, allowing laboratories to collect the right samples at the right time and under the right conditions [63].

## 10. Conclusions

Red blood cell distribution width is a newly recognized prognostic marker for a variety of illnesses, including viral infections. Additionally, RDW has strong specificities and sensitivities for predicting the severity and progression of hepatitis viruses. RDW is a cheap and easily accessible laboratory measure, indicating its likely enormous value in nations with limited resources compared to other biomarkers. However, there are still certain restrictions on the widespread application of RDW for viral illness prognostic prediction. To ensure the consistent use of RDW as a predictive biomarker for developing viral infections, additional research is required to overcome the current limitations. The whole literature data suggests that RDW could potentially be used as a prognostic and predictive marker for hepatic injury and inflammation. Some of the possible mechanisms are discussed above. Another proposed mechanism for liver injury could be increased intestinal iron absorption via the intestinal pathway and increased iron accumulation within Kupffer cells. This results in the formation of reactive oxygen species, causing lipid peroxidation, leading to cellular protein and DNA degradation, and causing upregulation and activation of hepatic stellate cells and smooth muscle actin, eventually causing hepatic fibrosis [64]. Although the exact mechanism of action remains unknown, a possible hypothesis could be linked to increased iron storage and impaired usage, leading to ineffective erythropoiesis and causing elevated RDW. Given its frequent use, it can be utilized as a marker in chronic liver disease. Furthermore, it can be combined with other traditional markers for predictive and prognostic purposes in chronic liver disease. Nevertheless, these results should be interpreted with caution as no classic literature is available, and hence, further detailed analysis and studies need to be performed to establish RDW significance as an inflammatory biomarker in chronic liver disease.

## Figures and Tables

**Figure 1 ijms-24-03487-f001:**
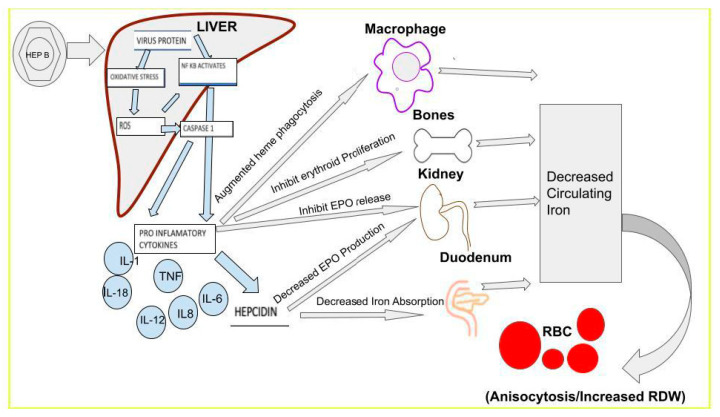
This figure represents chronic hepatitis B infection and its pathophysiology leading to changes in RDW from infecting the liver and causing the release of proinflammatory cytokines and their effects on different organ systems, eventually leading to altered RBCs leading to an increase in RDW.

**Figure 2 ijms-24-03487-f002:**
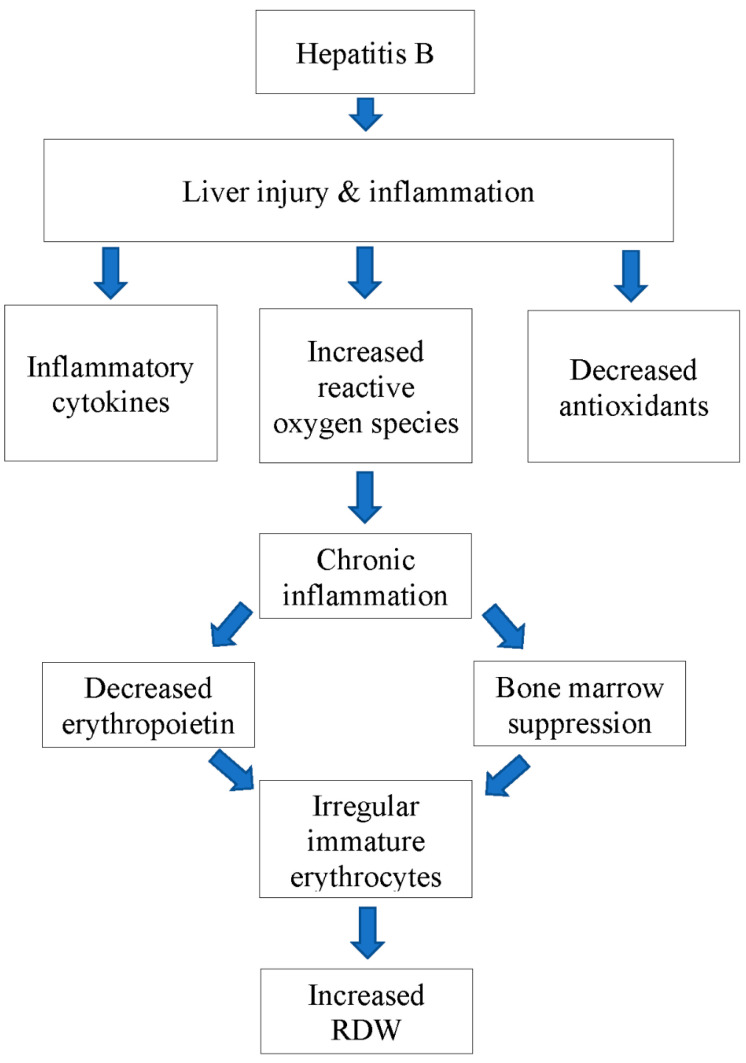
Flow chart depicting the possible mechanism of action of Chronic HepB infection causing increased RDW.

**Table 1 ijms-24-03487-t001:** Association of liver disease with RDW and its significance.

Disease	Author, Year, Study Type	Sample	Disease Stage	RDW	OR, CI, *p* Value
HepB	Lou, 2012, Retrospective [16]	171	Acute/Chronic HepB	18.3	NR, 15.19–21.41, <0.001
HepB	Chen B., 2013, Retrospective [17]	458	Liver Cirrhosis	NR	2.06, 1.21–3.51, <0.001
HepB	Karagoz E., 2014, Retrospective [18]	229	Hepatic Fibrosis	NR	1.803, 1.331–2.333, <0.001
HepB	Huang R., 2014, Retrospective [19]	130	Liver Cirrhosis	16.07	NR, 13.66–18.48, <0.01
HepB	Xu S.W., 2015, Retrospective [20]	519	Hepatic Fibrosis	NR	1.146, 1.008–1.303, 0.04
HepB	Wang H., 2016, Retrospective [21]	218	Liver Cirrhosis	14.5	NR, 13.0–15.4, <0.001
HepB	Fan X., 2018, Meta-analysis [22]	5481	Chronic HepB Infection	NR	1.309, 0.775–1.843, <0.001
HepC	He Q., 2016, Retrospective [23]	178	Liver Cirrhosis	16.44	NR, 13.43–19.45, <0.001
HepC	Nar R., 2016, Retrospective [24]	82	Liver Cirrhosis	15.05	NR, 13.17–16.93, <0.001
HepE	Jian W., 2019, Meta-analysis [25]	302	Liver Cirrhosis	14.2	NR, NR, <0.05
NAFLD	Cengiz M., 2013, Retrospective [26]	62	Hepatic Fibrosis	14.28	1.73, 1.242–2.409, <0.01
NAFLD	Yang W., 2014, Observational [27]	619	NAFLD	13.23	1.167, 1.052–1.294, <0.004
AIH	Zeng T., 2018, Retrospective [28]	76	Liver Cirrhosis	17.6	1.474, 1.086–1.842, <0.001
AIH	Wang H., 2019, Retrospective [29]	92	AIH secondary to HepB	15.6	1.702, 0.634–0.843, <0.001
PBC	Wang H., 2016, Retrospective [30]	94	Liver Cirrhosis	14.4	NR, 13.3–15.3, <0.019

Note: RDW (red cell distribution width), HepB (hepatitis B), HepC (hepatitis C), HepE (hepatitis E), NAFLD (non-alcoholic fatty liver disease), AIH (autoimmune hepatitis), PBC (primary biliary cirrhosis), OR (odds ratio), and CI (confidence interval); NR—not reported.

## Data Availability

Not applicable.

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
