# Peer review of "The Role of Red Cell Distribution Width as a Prognostic Marker in Chronic Liver Disease: A Literature Review"

_ijms, 2023, doi:10.3390/ijms24043487_

Round 1

Reviewer 1 Report

General comments:

The review of Aslam et al. concerns the discussion on the possible use of the RDW value as potential marker in the prognosis of several diseases. The subject of this review, in my opinion, is surely of interest for the scientific audience, however in its current form this manuscript is far from being suitable for publication in IJMS.

My consideration arises from several critical aspects of this review:

1) the cited literature is not very actual, the great majority of the papers cited are from six to ten years old, for this reason this review does not represent the very actual scientific panorama on the field. Just to give some examples, (i) the final conclusion of the review is based a paper from 2008, (ii) the data on the spread of the pathologies are outdated, (iii) at the first line of the paper the values reported are from 2018.  The authors should carefully check the more recent works available in the literature in order to provide a more actual view of the use of RDW

2) the writing is difficult to understand, in some cases are missed subjects and verbs. The authors should perform an extensive editing of the entire manuscript. This difficulty arises already in the abstract, it looks like a cut and paste of phrases from the review instead of an homogeneous discussion.

3) the table number 1 is very superficial, written in a very bad English and without citations. It should be completely re-invented.

4) the table number 2 is very difficult to understand and somehow misleading. If the work is focused on the values of RDW why there is a consistent number of entries in the table in which this value is NA?? 

5) The general impression on the review is that it was done in a very superficial way, for example: (i) the RWD values from the cited papers are simply reported in a row in the text without graphs histograms in order to make them more understandable, (ii) the discussion on the mechanisms underlying the different pathologies in really superficial, main reason that makes the review "short"; (iii) the figures are really basic and too simple, without adding information to the text, moreover they regards only the mechanisms and not the RWD that is the subject of the review; (iv) the list of authors contains several mistakes in the names and affiliations; (v) some citations are from old internet pages instead of scientific publications or books; (vi) the future directions paragraph, in my opinion, should be the real added value to the review is simply too short while the rest of the paper is a list of results from the literature with very few comments and discussions.

For these reasons, in my opinion, this review is not suitable for publication in IJMS in its actual form. It should be completely re-written and implemented with more recent results and discussions/considerations by the authors.

Author Response

Thank you very much for taking the time to review our manuscript and provide suggestions to improve it. We believe that after the revision has improved substantially, and we hope the revised manuscript will be suitable for publication. Please see the attachment

Reviewer 2 Report

This review article is thought provoking.   It explores the literature on six chronic liver diseases that correlate disease progression indicators to Red Cell Distribution Width (RDW).  In six well organized sections, the odds ratio, confidence interval and p-value are summarized from each study.  The review also includes two tables.  The first table outlines, by liver disease type, the likely mechanism leading to the production of larger red cells. The overarching conclusion is that inflammation correlates with elevated RDW. Repeated calls for the need for further study are noted throughout the review.  

In contrast to the rest of the review, the conclusions were weak and poorly organized.  This section is key to drawing the parallels between each CLD reviewed and again make the case for the importance of mechanistic studies. Instead, a new mechanism is proposed that is tangentially related to one mechanism in Table 1.  This section needs content improvements to strengthen it. 

Editorial suggestions to improve readability of the manuscript include: 

·       Text in figure 1 is inconsistent font and size.  Capitalizations are inconsistent.  Minimum font size should be increased to 10 or greater in the final figure size.  

·       The word Association is capitalized in unusual locations in the manuscript.

·       Line 320.  Sentence structure when referencing Zeng needs work.

·       Line 360.  “It” is not clearly defined in the sentence.

·       Line 384. COX is not defined.

In summary, this work outlines gaps in knowledge with the message that further research is needed to understand mechanisms leading to increased RDW.   The manuscript fills a need in the field by drawing parallels between findings from six different CLD.  I recommend minor edits based upon the items noted above.  

Author Response

Thank you very much for taking the time to review our manuscript and provide suggestions to improve it. We believe the revision has improved the manuscript substantially, and we hope the revised manuscript will be suitable for publication. Please see the attachment

Reviewer 3 Report

RBC size distribution is a routine hematological parameter. Its clinical and diagnostic significance was studied for many decades. This parameter (e.g., shift to the right) may be changed by many biological factors.

The authors propose the existing test panel for assessment of liver diseases, concluding on potential usage of increased RBC distribution width (RDW) values as a prognostic and predictive marker for hepatic injury and inflammation. The authors suggest that it may be combined with traditional inflammatory markers for predictive and prognostic purposes in chronic liver disease.

In fact, this standard hematological parameter is suggested to be a surrogate prognostic/predictive marker of liver disease progression, however, in combination with basic clinical and laboratory data.

Remarks:

Lines 84-95, Table 1: The multiple factors of mean RBC size changes are briefly listed here, e.g., impaired erythropoiesis, Hb synthesis, liver and gut metabolism, iron turnover, vitamin B12 supply and production etc. This useful review shows a variety of potential metabolic effects on RBC size in the patients with liver disease.  

Table 2: The presented data mainly concern associations between clinical stage of hepatitis B, and mean RBC size. However, prognostic or predictive value of some parameter implies, at least, its correlation with later clinical progression, e.g., at >1 years in prospective study. To this item, the authors should present more prospective studies from the literature, or point to their necessity in future studies.

Sections 2-8: In the following text, the authors discuss probable associations between RDW changes and different hepatic disorders and their clinical stages. In fact, these associations are considered a good indirect parameter for clinical characterization.

However, implementation of RDW markers in hepatology strictly depends on assessment of its normal ranges in various populations, and cut-off values for practical usage. This caution should be stressed in conclusions.

Author Response

Thank you very much for taking the time to review our manuscript and provide suggestions to improve it. We believe the revision has improved the manuscript substantially, and we hope the revised manuscript will be suitable for publication. Please see the attachment.

Round 2

Reviewer 1 Report

In this second version the authors had performed a complete restyle and rewriting of their manuscript,

all my previous doubts and concerns had been addressed.

There is only one suggestion that regards the use of figure 2: actually, it is not even cited in the paper, and, in my opinion, it is not even necessary and can be removed from the manuscript. 

Once the figure 2 is removed (or it is properly cited in the text), in my opinion the manuscript will be suitable for publication in IJMS.

Reviewer 3 Report

The authors have made a sufficient work to clear the questions posed in the review. E.g., they added extended characteristics of different types of hepatitis. In Discussion and Conclusions they consider the issues of specificity in more cautious manner and stress a need for additional studies of RCDW as a marker of clinical dynamics in distinct cases. The article may be published in current version.